# Effects of COMT Genotypes on Working Memory Performance in Fibromyalgia Patients

**DOI:** 10.3390/jcm9082479

**Published:** 2020-08-01

**Authors:** David Ferrera, Francisco Gómez-Esquer, Irene Peláez, Paloma Barjola, Roberto Fernandes-Magalhaes, Alberto Carpio, María E. De Lahoz, Gema Díaz-Gil, Francisco Mercado

**Affiliations:** 1Department of Psychology, Faculty of Health Sciences, Rey Juan Carlos University, 28922 Madrid, Spain; david.ferrera@urjc.es (D.F.); irene.pelaez@urjc.es (I.P.); paloma.barjola@urjc.es (P.B.); roberto.fernandes@urjc.es (R.F.-M.); alberto.carpio@urjc.es (A.C.); me.delahoz.2017@alumnos.urjc.es (M.E.D.L.); 2Department of Basic Health Sciences, Faculty of Health Sciences, Rey Juan Carlos University, 28922 Madrid, Spain; francisco.gomez.esquer@urjc.es (F.G.-E.); gema.diaz@urjc.es (G.D.-G.)

**Keywords:** fibromyalgia, COMT, working memory, verbal working memory, spatial working memory

## Abstract

Growing research has reported the presence of a clear impairment of working memory functioning in fibromyalgia. Although different genetic factors involving dopamine availability (i.e, the COMT gene) have been associated with the more severe presentation of key symptoms in fibromyalgia, scientific evidence regarding the influence of COMT genotypes on cognitive impairment in these patients is still lacking. To this end, 167 participants took part in the present investigation. Working memory performance was assessed by the application of the SST (Spatial Span Test) and LNST (Letter and Number Sequence Test) belonging to the Weschler Memory Scale III. Significant working memory impairment was shown by the fibromyalgia patients. Remarkably, our results suggest that performance according to different working memory measures might be influenced by different genotypes of the COMT gene. Specifically, fibromyalgia patients carrying the Val/Val genotype exhibited significantly worse outcomes for the span of SST backward, SST backward score, SST total score and the Working Memory Index (WMI) than the Val/Val healthy carriers. Furthermore, the Val/Val patients performed worse on the SST backward and SST score than heterozygotes. Our findings are the first to show a link between the COMT gene and working memory dysfunction in fibromyalgia, supporting the idea that higher COMT enzyme activity would contribute to more severe working memory impairment in fibromyalgia.

## 1. Introduction

Neurobiological evidence gathered from the past decade has established that dopamine plays a crucial role in neurotransmission mechanisms supporting mental operations involved in working memory processing [1,2,3,4,5,6,7,8,9]. In line with this, whereas dopamine depletion in the prefrontal cortex (PFC) has been related to the appearance of deficits in working memory [10,11,12], the administration of dopamine agonist drugs (e.g., pergolide or bromocriptine) has been associated with an improvement in such processes [13,14,15,16,17].

Catechol-O-methyltransferase (COMT) is an enzyme involved in the breakdown of several catecholamine neurotransmitters, such as dopamine, noradrenaline and serotonin [18,19,20]. Specifically, it has been reported that this enzyme also regulates the amount of dopamine in the PFC [20,21], amygdala, entorhinal cortex or hippocampus [22,23,24]—regions belonging to the neural network underlying working memory processing. One of the most studied single nucleotide polymorphisms (SNPs) of the COMT gene is the rs4680 polymorphism, which arises by a substitution, in the 158 codon, from valine (Val) to methionine (Met) [25]. This change causes the enzyme encoded by the gene to have about 1/4 less activity [26], influencing, in turn, the availability of dopamine within the dorsolateral prefrontal cortex (DLPFC) [27]. Several experimental studies have demonstrated that performance in working memory tasks may differ as a consequence of carrying different alleles of this gene [28,29,30,31,32,33,34]. It has been reported that those subjects carrying Met alleles showed better functioning in verbal and visuo-spatial working memory tasks [28,30,34], and fewer perseverative errors on the Wisconsin Card Sorting Test (WCST) than Val carriers [35,36]. These data also seem to have a neurophysiological correlate, since brain imaging studies have shown that Val carriers exhibited a decrease in dopamine levels within the PFC, reducing the blood-oxygen-level dependent (BOLD) response of this brain region compared to that in the Met carriers [31,36], and also reflected by the poor performance of Val carriers in working memory tests [29]. These results have also been replicated in chronic patients, such as those suffering from posttraumatic stress disorder or schizophrenia [36,37,38,39], leading to the consideration of this gene as a therapeutic target for improving cognitive symptoms [40].

The COMT gene has been widely studied with respect to the pain-related and affective symptoms of chronic pain syndromes, such as fibromyalgia [41,42,43,44]. Experimental evidence has suggested that fibromyalgia patients carrying the two Met alleles of the COMT gene had a higher number of tender points [42], greater sensibility to painful stimulation [41,43] and, also, a higher severity of depressive and anxious symptomatology than Val carriers [43,44,45].

Along with affective and physical symptoms, subjective complaints about cognitive functioning (encompassed under the term fibrofog [46]) are one of the main symptoms in fibromyalgia [47,48]. The cognitive impairments have been proposed as more disabling than the pain itself [49]. Systematically, neuropsychological evidence has shown that fibromyalgia patients exhibit significant cognitive dysfunction compared to healthy control participants in different domains, such as attention [50,51,52], processing speed [53], executive functioning [54,55,56,57,58] and theory of mind [59]. Remarkably, it has been suggested that these impairments might be better explained by alterations in working memory processes [60,61], where these findings were particularly solid [62,63].

However, despite the frequently reported relationship between working memory impairment and fibromyalgia, the effect of biological indices (i.e., the COMT gene) related to dopamine availability on cognitive resources in these patients has not been explored to date. Previous experimental evidence coming from voxel-based morphometry studies [64] has shown clear decrements in cerebral grey matter density in fibromyalgia patients within regions related to working memory processing, such as the hippocampal gyrus and the anterior cingulate cortex (ACC) [65,66]. Furthermore, these changes in cerebral grey matter have been linked to the presence of dopamine metabolism abnormalities in fibromyalgia [64]. In line with these findings, some authors have recently pointed out the need to explore the relationship between dopamine and cognitive processes in chronic pain [67]. This accumulated evidence leads us to suggest that COMT, as a genetic marker involved in dopamine availability, might be a useful target to better characterise cognitive dysfunction in fibromyalgia.

In summary, this investigation aimed to explore the relationship between the different genotypes of COMT and working memory dysfunction in fibromyalgia. To this end, we analysed the rs4680 SNP of the COMT gene in fibromyalgia patients and healthy participants and measured their performance in neuropsychological tests involving working memory processing. Based on previous findings, we expected to confirm the presence of cognitive impairment in fibromyalgia. Additionally, we hypothesised that this altered neuropsychological functioning regarding working memory processing would be even worse in those patients carrying the Val/Val genotype of the SNP rs4680 than in the Met allele carriers.

## 2. Materials and Methods

### 2.1. Sample Size Calculation

In order to calculate the number of participants required for this study, the G*Power software was used [68]. This software allows one to determine the sample size necessary to obtain a certain statistical power. In this study, the sample size was calculated using the a priori statistical power of 1 − β = 0.80 for a medium effect size. The total sample size obtained for this statistical power was 158 subjects. We expected that 10% of participants’ data would be lost during the experimental procedure; therefore, we decided to increase the total sample size to 173. Data from six subjects were lost because they did not complete all the tests; thus, one hundred and sixty-seven Caucasian women took part in the experiment. Eighty-six were fibromyalgia patients, and eighty-one were healthy control participants (HC).

### 2.2. Participants

Fibromyalgia patients were recruited from the Fibromyalgia and Chronic Fatigue Syndrome Association (AFYSYFACRO) and Fibromyalgia Association of Pinto (AFAP) of Comunidad de Madrid (Spain). All fulfilled the diagnosis criteria proposed by the American College of Rheumatology (ACR) in 2010 [47]. The diagnosis of fibromyalgia was carried out by different rheumatologists among different public hospitals of the Comunidad de Madrid. HC participants were recruited by email, by advertisements published along the Faculty of Health Science at the Rey Juan Carlos University and among the friends of the fibromyalgia patients. The whole sample was aged between 35 and 67 years old. The HC participants were selected in such a way that the age and educational level were matched with those of the patients. The HC participants did not suffer from any chronic pain conditions. All participants had normal or corrected-to-normal eyesight and the ability to read and write in Spanish at the equivalent of eighth grade level in English. No differences were found in terms of age (F _(1165)_ = 0.001, *p* = 0.975) or educational level (χ^2^
_(3)_ = 3.305, *p* = 0.347) between the two groups of participants. They had no history of psychiatric disorders (i.e., psychosis, personality disorders or substance abuse/dependence) or neurological diseases. The HC participants did not suffer from any chronic pain conditions. Patients who took medications kept doing so for both medical prescription and ethical considerations. Most patients were taking analgesics, antidepressants and benzodiazepines.

### 2.3. Procedure

Data for whole sample were collected in the Faculty of Health Science of Rey Juan Carlos University (Madrid). Once in the laboratory, participants gave written informed consent for their involvement in the experiment. Subsequently, we collected the demographic data, clinical information and drug consumption information. Once all this information was collected, the neuropsychological evaluation was always carried out in the same order by an expert psychologist. Finally, saliva samples were collected according to the manufacturer’s protocol from all participants. Each session lasted approximately 45 to 60 min.

### 2.4. Self-Reported Measurements and Psychological Pain Assessment

Participants provided written informed consent for their involvement in the experiment. The Rey Juan Carlos Research Ethics Board approved this study (ref: 201000100011550), which followed all the requirements of this committee. Subsequently, participants completed some self-report instruments in their validated Spanish version. The whole sample fulfilled the State-Trait Anxiety Inventory (STAI) [69]. The STAI test is a self-report questionnaire composed of 20 items for measuring Trait Anxiety and 20 items for State Anxiety ranging from 0 to 60 (subsequently assigning a centile score). The reported values of Cronbach’s α were 0.93 for the Trait Anxiety and 0.87 for State Anxiety subscales [69]. The Beck Depression Inventory (BDI) [70] was also administered to all participants for measuring depressive symptoms. This scale has scores ranging from 0 to 63. The values of Cronbach’s α were 0.87 [70]. Additionally, fibromyalgia patients completed the Spanish version of the Fibromyalgia Impact Questionnaire (FIQ-S) [71], a specific questionnaire to assess the current health and functional status of fibromyalgia patients with a score ranging between 0 and 80. Just before starting the experiment, a visual analogue scale (VAS) ranging from 0 (no pain at all) to 10 (worse imaginable pain) was applied for monitoring the level of pain suffered by the participants in the previous week, including the day of testing. The reported Cronbach’s α for this test was 0.82 [71]. The fibromyalgia patients showed higher scores for depression (F _(1165)_ = 111.431, *p* = 0.001, η^2^_p_ = 0.403), state anxiety (F _(1165)_ = 11.534, *p* = 0.001, η^2^_p_ = 0.065), trait anxiety (F _(1165)_ = 51.981, *p* = 0.001, η^2^_p_ = 0.240) and VAS pain (F _(1165)_ = 329.208, *p* = 0.001, η^2^_p_ = 0.666) than the control participants. Data regarding the psychological variables are shown in Table 1, along with the information on drug consumption and sociodemographic conditions.

### 2.5. Neuropsychological Testing of Working Memory

Working memory functioning was assessed by two different neuropsychological tests. Both tests are part of the Working Memory Index (WMI) belonging to the Spanish version of the Wechsler Memory Scale III (WMS-III) [72]. This index allows one to obtain a measure of the individual’s performance in different tasks that require attending to, retaining and processing information in memory, and subsequently generating a response according to that information [72]. The WMI is calculated from the scores obtained in the Spatial Span Test (SST) and the Letter and Number Sequence Test (LNST). These tests were selected because they have been considered as validated and standardized measures of verbal and visual–spatial functions of working memory functioning. Values of Cronbach’s α between 0.85 and 0.99 were reported [73].

The SST of the WMS-III evaluates immediate spatial working memory [74]. This task is a modified version of the classical Corsi Block-Tapping Test [75] and has been previously used in chronic patients with working memory impairments [76,77]. Participants were presented with a board on which a set of ten cubes or blocks were fixed. On the board, all the cubes had written on them numbers from 1 to 10 that only the examiner could see. The SST test was applied twice, one forwards and the other backwards (see Figure 1). In the forward application, participants were asked to repeat in the same order the tapping sequence previously made by the examiner. The first sequence presented to the participants started with a tapping sequence of three blocks. Two trials were applied for each sequence length. The SST finished when the participants failed to reproduce both trials of the same sequence. For the backward application, participants were asked to repeat the tapping sequence but in reverse order (i.e., backwards). Whereas performing the forward version requires the setting in motion of several processes such as encoding, memory delay and the implementation of response alternatives, the backward application requires the activation of executive processes such as manipulating information [78]. Performance in this test was measured by recording the following raw scores: span of SST forward (number of correct trials performed in the forward application; score ranging from 0 to 9), SST forward score (sum of all the points obtained by the subject in direct order with a score ranging from 0 to 16), span of SST backward (number of correct trials performed in the backward application; in the same way as in the forward application, the score range was 0 to 9), SST backward score (sum of all the points obtained by the subject in the backward application with a score ranging from 0 to 16) and SST total score (sum of SST forward score and SST backward score, whose range of scores was 0 to 32).

The LNST was used for assessing verbal working memory. A series of sequences consisting of disordered letters and numbers were verbally presented by the examiner to the participants, who were instructed to reproduce them in a specific order: first, the numbers from the smallest to the largest, followed by the letters in alphabetic order. The lengths of the LNST series were gradually increased, starting from lengths of two elements and finishing with series including eight elements (e.g., two elements: L-2 (2-L); eight elements: 7-M-2-T-6-F-9-A (2-6-7-9-A-F-M-T). The same criterion applied in the SST test for stopping its application was also adopted for the LNST test (i.e., it finished when the participants failed two consecutive elements belonging to the same sequence length). This scale has been shown to be sensitive for exploring working memory disorders in different chronic syndromes [79,80]. Two different memory measures were computed for this test: the span of LNST (maximum number of elements correctly remembered; score ranging from 0 to 8) and LNST score (sum of all the points obtained by the subject in this task; score ranging from 0 to 21). Finally, the total score of the WMI (score ranging between 50 and 150) was also calculated by computing the total scores of the LNST and SST tests.

### 2.6. COMT Genotyping

Unstimulated whole saliva samples were collected into collection tubes according to standardized procedures. Subjects were asked not to eat, drink or chew gum for 1 hour before sample collection. Immediately after collection, the samples were stored at −20 °C until analysis. Saliva, as compared to blood collection, has the following advantages: it requires no specialized personnel for collection, allows for remote collection by the patient, is painless, is well accepted by participants, has decreased risks of disease transmission, does not clot, can be frozen before DNA extraction and possibly has a longer storage time.

Genomic DNA was extracted from 5 mL of saliva using a REALPURE Saliva RBMEG06 Kit (Durviz, Valencia, Spain) according to the manufacturer’s protocol. The resulting DNA was diluted to 100–1000 ng/μL, using 1×Tris-EDTA (TE) buffer (Sigma-Aldrich, Dorset, UK) and assessed for purity and concentration using a NanoDrop™ ND1000 Spectrophotometer (Thermo Fisher Scientific Inc., Hemel Hempstead, Hertfordshire, UK). COMT polymorphisms were genotyped by real-time polymerase chain reaction (RT-PCR) analysis using TaqMan® Predesigned SNP Genotyping Assays for rs4680 polymorphisms (Thermo Fisher Scientific Inc., Hemel Hempstead, Hertfordshire, UK). TaqMan® SNP Genotyping Assays use TaqMan® 5′-nuclease chemistry for amplifying and detecting specific polymorphisms in purified genomic DNA samples. Each assay allows the genotyping of individuals for a single nucleotide polymorphism (SNP). Each TaqMan® SNP Genotyping Assay contains: (A) sequence-specific forward and reverse primers to amplify the polymorphic sequence of interest and (B) two TaqMan® minor groove binder (MGB) probes with non-fluorescent quenchers (NFQ): one VIC^TM^-labeled probe to detect Allele 1’s sequence and one FAM^TM^-labelled probe to detect Allele 2’s sequence. Amplification was carried out an ABI Prism 7000 Sequence Detection System (Thermo Fisher Scientific Inc., Hemel Hempstead, Hertfordshire, UK) in the Genomics unit of the Technological Support Center (CAT) of the Rey Juan Carlos University. All genotypes were determined twice.

### 2.7. Data Analysis

The normal distribution of the dependent variables was checked using the ratio of kurtosis and asymmetry to the standard error. This index for each variable was between +2 and −2, confirming that the scores did not violate the assumption of normality [81,82].

A chi-square test (χ²) was used to evaluate the statistical distribution of the COMT genotypes in both the fibromyalgia patients and HC participants in order to verify the Hardy–Weinberg equilibrium (HWE).

To explore the potential effect of the COMT rs4680 polymorphism (participants were grouped in three different genotypes: Met/Met, Val/Val and Met/Val) on the functioning of working memory, two-way ANOVAs were performed. The span of SST forward, SST forward score, span of SST backward, SST backward score, SST total score, span of LNST, LNST score and WMI were considered as dependent variables, while the COMT genotypes (Met/Met, Val/Val and Met/Val) and groups (two levels: fibromyalgia and HC participants) were set as the between-subject factors. Post hoc comparisons to determine the significance of pairwise contrasts were performed using Bonferroni adjustment (α = 0.05) for controlling the Type I error rate. Effect sizes were computed using the partial eta-square (ƞ^2^_p_) method only for the significant analyses. We also estimated statistical power using the observed power (1 − β), only for the significant interactions.

Additionally, the association between the clinical questionnaire scores and the performance in the working memory tests was explored in two ways. First, at an exploratory level, Pearson correlations (r) were determined. Secondly, to test whether the effects in the working memory tests were strictly due to the COMT genotypes but not influenced by the differences in the clinical scores between groups, a series of ANCOVAs were performed. Thus, those clinical variables that were significantly associated with neuropsychological outcomes were introduced in these analyses as covariables (i.e., depression, anxiety and pain) in order to neutralise their possible influence on the statistical results. Effect sizes were computed using the partial eta-square (η^2^_p_) method. A significance level of 0.05 (two tailed) was used for all statistical analyses where significant.

As has been previously reported, benzodiazepines and antidepressants might have different effects on cognition [83]. To account for this, several control analyses were performed via ANOVAs on working memory scores, only including fibromyalgia patients using and not using particular medications. We introduced the psychotropic drugs (benzodiazepines and antidepressants) as factors, whereas the working memory measures were dependent variables. All statistical analyses were performed with the SPSS package (v.25.0; SPSS Inc., Chicago, IL, USA).

## 3. Results

### 3.1. COMT Polymorphism Frequencies

The chi-square (χ²) test was used to assess the distribution of the genotypes in order to check the HWE. The frequency of the COMT rs4680 polymorphism distribution fulfilled the HWE in both the HC participants (χ^2^ = 0.351, *p* = 0.553) and fibromyalgia patients (χ^2^ = 0.04, *p* = 0.825). Statistical data related to the genotype and allele frequency distributions of the COMT gene considering each group of participants are displayed in the Table 2.

### 3.2. Statistical Effects in the Neuropsychological Assessment

ANOVAs revealed a major effect of group, where fibromyalgia patients showed significantly lower scores than HC participants in most of the working memory measures: LNST score (F _(1161)_ = 6.603, *p* = 0.011, η^2^_p_ = 0.039), span of SST forward (F _(1161)_ = 6.712, *p* = 0.010, η^2^_p_ = 0.040), SST forward score (F _(1161)_ = 6.495, *p* = 0.012, η^2^_p_ = 0.039), span of SST backwards (F _(1161)_ = 5.343, *p* = 0.022, η^2^_p_ = 0.032), SST backward score (F _(1161)_ = 6.618 *p* = 0.011, η^2^_p_ = 0.039), SST total score (F _(1161)_ = 9.080, *p* = 0.003, η^2^_p_ = 0.053) and WMI (F _(1161)_ = 9.709, *p* = 0.002, η^2^_p_ = 0.057). Only scores related to the span of LNST did not yield significant differences between both groups (F _(1161)_ = 1.126, *p* = 0.327).

### 3.3. Effects of Interaction between COMT Genotypes and Group of Participants on Working Memory Performance

The mean scores for both the verbal and spatial working memory tests related to each group of participants are displayed in Table 3. Regarding the interaction between the COMT genotypes and group, the ANOVAs showed statistically significant differences for the SST backward score (F _(2161)_ = 5.543, *p* = 0.005, η^2^_p_ = 0.064, 1 − β = 0.849) and SST total score (F _(2161)_ = 4.921, *p* = 0.008, η^2^_p_ = 0.058, 1 − β = 0.801). In both cases, post hoc analyses showed a similar difference pattern. Specifically, fibromyalgia patients carrying the Val/Val genotype showed significantly lower scores than the Val/Val genotype carriers belonging to the HC group (*p* = 0.001 for the comparison of the SST backward score and SST total score), as can be also seen in Figure 2. We also found a significant difference for the fibromyalgia group between those patients carrying the Val/Val and Met/Val genotypes (*p* = 0.03 for the SST backward score and *p* = 0.014 for the SST total score). Specifically, the Val/Val genotype carriers performed worse according to both neuropsychological measures than the patients carrying other genotypes. In addition, an interaction effect between the COMT genotype and group was also found for the span of SST backward score (F _(2161)_ = 4.013, *p* = 0.020, η^2^_p_ = 0.047, 1 − β = 0.710). Post hoc analyses revealed a significant difference between the fibromyalgia and HC groups that only was true for those participants carrying the Val/Val genotype (*p* = 0.001). In this case, the fibromyalgia patients showed a lower performance than the HC participants (see Figure 2). Finally, we found a statistically significant interaction between the COMT genotype and group for the WMI (F _(2161)_ = 4.095, *p* = 0.018, η^2^_p_ = 0.048, 1 − β = 0.720). Post hoc analyses showed that the fibromyalgia group scored lower than the HC participants. This difference was obtained for both the Val/Val genotype (*p* = 0.007) and Met/Met genotype (*p* = 0.014) carriers (Table 3). No other comparison yielded significant results for the span of LNST, LNST score, span of SST forward and SST forward score. In addition, no major effect of COMT was found.

### 3.4. Relationship between the Clinical Variables and Neuropsychological Test Results

As mentioned in the Data Analysis section, we evaluated the relationship between the scores obtained from the clinical questionnaires and the working memory test results in terms of Pearson correlations (see Table 4). Trait anxiety (Trait-STAI) correlated negatively with the number of correct trials performed in the Spatial Span test (Span of SST forward) (*r* = −0.219, *p* = 0.004), the number of correct backward trials performed in the Spatial Span test (Span of SST backward) (*r* = −0.206, *p* = 0.008), the score for the Spatial Span test in the forward application (SST forward score) (*r* = −0.209, *p* = 0.007) and the total score for the Spatial Span test (SST total score) (*r* = −0.192, *p* = 0.013). In other words, the higher the scores related to trait anxiety were, the lower the span of SST forward, the span of SST backward, the SST forward score and the SST total score were. Moreover, depression (BDI) and the Letter and Number Sequence test score (LNST score) (*r* = −0.157, *p* = 0.042) showed a significant association. These variables were also inversely correlated, so the higher the depression, the lower the LNST score. Additionally, depression and the Working Memory Index (WMI) showed a significant association (*r* = −0.64, *p* = 0.035); the lower the WMI, the higher the scores related to depression. Finally, pain outcomes correlated with the number of correct trials performed in the direct order of the Spatial Span test (span of SST forward) (*r* = −0.157, *p* = 0.043). The rest of the analyses did not show statistical differences, as can be seen in Table 4.

Based on the results provided by the correlation analyses, ANCOVAs were carried out including the clinical variables showing a significant association with the working memory test scores as covariables. The analyses revealed that the differences shown in the working memory measures as a function of the interaction between the COMT genotypes and group were independent of the clinical symptomatology. Specifically, the COMT genotype and group interactions, after controlling for scores in trait anxiety, remained significant for the span of SST backward (F _(2160)_ = 3.827, *p* = 0.024, η^2^_p_ = 0.046) and SST total score (F _(2160)_ = 4.663, *p* = 0.011, η^2^_p_ = 0.055). Furthermore, the interaction effect between the COMT genotype and group for the WMI after controlling for depression scores also remained significant (F _(2160)_ = 4.051, *p* = 0.019, η^2^_p_ = 0.048). In the case of the rest of the significant variables described in the correlation analyses, they consistently displayed non-significant differences for the COMT genotypes and group interaction, when trait anxiety (span of SST forward (F _(2160)_ = 0.790, *p* = 0.456) and SST forward score (F _(2160)_ = 1.606, *p* = 0.204)), depression (LNST score (F _(2160)_ = 1.593, *p* = 0.207)) or pain were introduced as covariables (span of SST forward (F _(2160)_ = 0.885, *p* = 0.415)).

### 3.5. Working Memory Analyses Controlling for the Intake of Psychotropic Drugs

As we have previously indicated, fibromyalgia patients usually have a high consumption of medications due to their chronic condition. Some of these drugs, such as antidepressants and benzodiazepines, can have a negative impact on cognitive performance. Therefore, we carried out several control analyses in order to explore the potential effects of drug consumption on cognitive functioning within the fibromyalgia group. ANOVAs were performed for patients who were taking and were not taking benzodiazepines or antidepressant drugs. The analyses revealed that working memory outcomes were not significantly influenced by the medications taken for fibromyalgia patients (see Table 5).

## 4. Discussion

The present investigation aimed to explore the relationship between the different genotypes of the COMT gene and working memory dysfunction in fibromyalgia. The results derived from the statistical analyses showed that patients with fibromyalgia had significantly lower scores on both the Letter and Number Sequence and Spatial Span tests. More interestingly, our findings suggest that the performance according to different working memory measures might be modulated by the different genotypes of the COMT rs4680 polymorphism. Specifically, fibromyalgia patients carrying the Val/Val genotype (i.e., having high COMT activity leading to higher dopamine degradation) exhibited significantly worse performance in terms of the span of SST backward, SST backward score, SST total score and Working Memory Index of the WMS-III. To the best of our knowledge, this is the first study that establishes a relationship between COMT genotype and working memory dysfunction in fibromyalgia.

As indicated in the Results section, our data have confirmed the presence of clear working memory dysfunction in fibromyalgia as reflected by the low performance according to the different neuropsychological measures (LNST and SST). Although there is some controversy about the nature and extent of the cognitive impairment in fibromyalgia [84,85,86,87], a wide group of studies using varied methodologies—such as neuropsychological tests [55,57,59,63,88,89], brain electrophysiological response assessments [54,58] and neuroimaging techniques [90,91,92]—has evidenced that cognitive functioning differs between fibromyalgia patients and healthy controls. In this sense, the available data suggest that cognitive impairment in fibromyalgia could be more evident when cognitive demands are elevated, as occurs in working memory tasks, where patients have to deal with situations that involve stimulus competition or require the allocation of cognitive resources to process distinct kinds of information at the same time [53,92,93].

What is more interesting regarding the current findings is the potential role of the COMT gene as a factor contributing to the cognitive impairment in fibromyalgia. Remarkably, our data have indicated that those patients carrying the Val/Val genotype of COMT exhibited significantly worse outcomes in terms of the span of SST backward, SST backward score, SST total score and the WMI than the HC participants carrying the same COMT genotype. Furthermore, we observed that homozygous valine fibromyalgia patients performed significantly lower in terms of the SST backward and SST total score than heterozygous patients. Finally, contrary to our expectations, the Met/Met genotype (i.e., that with low enzyme activity) was not associated with an improvement in working memory tasks in fibromyalgia. Although there is a lack of previous studies linking these genetic factors with cognitive dysfunction in fibromyalgia, the relationship between the COMT rs4680 polymorphism and working memory functioning has been consistently reported in healthy volunteers [28,30,31,34,94] and other pathologies characterized by working memory impairments such as schizophrenia or posttraumatic stress disorder [36,37,38,39]. It has been demonstrated that Val/Val carriers scored systematically lower than individuals carrying other genotypes on various working memory tasks. These chronic diseases have been characterized by brain activity abnormalities in prefrontal cortex (PFC) functioning while patients were involved in working memory tasks [95,96,97]. This scientific evidence is in accordance with the results derived from a growing body of investigations indicating that the PFC is a key brain region involved in the working memory impairments in fibromyalgia [90,91,92,98,99]. In this regard, Luerding and colleagues [90] have reported, using voxel-based morphometry, that fibromyalgia patients with lower grey matter volume in the left dorsolateral prefrontal cortex (DLPFC) had poorer performance in the Corsi Block-Tapping Test, confirming the crucial role of prefrontal brain regions for this working memory dysfunction. Furthermore, the variation in the dopamine levels in the PFC seems to be related to the variable neural response of this brain region during the performance of working memory tasks [33,36,100]. It has been suggested that the different COMT enzymatic activity seems to modulate the amount of available dopamine, exerting an effect, in turn, on the interindividual variability in cognition [101]. In this vein, a high enzymatic activity of COMT, as occurs in the Val/Val genotype carriers [26], would lead to diminished levels of dopamine at the synapse and less activity in postsynaptic receptors. Consequently, whereas this extracellular reduction in dopamine availability would be related to a worsening of working memory processes, the lower activity of the COMT enzyme (i.e., the Met/Met genotype) seems to contribute to the augmentation of dopamine levels at the synapse and, in turn, to higher working memory efficiency [6]. Thus, it has been proposed that the increase in extracellular dopamine could improve cognitive symptoms in fibromyalgia [102], due to the involvement of the dopaminergic system in cognitive impairment in this chronic pain syndrome [103]. Nevertheless, it is important to note that the optimal execution in working memory tasks would require intermediate dopamine levels (very high or low levels of dopamine would have negative effects on working memory performance) [7], since the relationship between working memory and dopamine availability seems to follow an inverted-U shape [6,104]. This prior evidence could help us to shed light on some of the unexpected results obtained here, such as those shown by patients carrying the Met/Met COMT genotype who did not reach the highest working memory performance. It leads us to conjecture that patients in this subgroup could potentially have very high levels of dopamine, which might disadvantage them for obtaining better performance in working memory tests. Although these data should be confirmed in future studies, our results suggest that the enzymatic activity of COMT might modulate the cognitive performance of fibromyalgia patients, mainly in spatial working memory tasks.

The relationship between dopamine availability and working memory dysfunction in fibromyalgia deserves further consideration. The effects of enzymatic COMT activity on the cognitive performance in patients with fibromyalgia were observed mainly in those neuropsychological measures involving visuo-spatial working memory capabilities (i.e., the span of SST backward, SST backward score and SST total score) but not in those assessing verbal working memory processes (i.e., the span of LNST and LNST score). The question that now arises is why that effect was not homogeneously reflected across the whole set of applied working memory tests. The answer to this question may admit two complementary but non-exclusive explanations. Firstly, both the difficulty and the cognitive load conveyed by a given task are relevant factors for detecting impairments in working memory functioning [92,105,106,107]. In this sense, the cognitive load generated by the two different working memory tests used here, the LNST and SST, seems to be different. It has been consistently reported that verbal and spatial working memory tasks can be separable for involving different sets of resources [108,109,110]. With respect to this, some studies have indicated that maintenance sub-processes might produce a similar cognitive load in both verbal and spatial working memory tasks, but updating sub-processes (which implies adding or removing representations in working memory (see [111])) could be qualitatively different in both tasks [112]. These authors argued that the irrelevant items included in a verbal working memory updating sequence caused higher proactive interference than those included in spatial working memory, as the semantic content of verbal tasks can create more durable representations in working memory than spatial stimuli. On the other hand, certain spatial tasks, such as the Spatial Span Test, require not only the implementation of working memory sub-processes (i.e., updating, encoding or maintenance) but also the coordination of motor planning processes and integration of spatial information [113]. This integration is not needed in verbal working memory tasks. Secondly, it has been reported that the neural circuit that connects the PFC and the hippocampus is a crucial pathway in spatial working memory tasks and this circuit could be modulated by cortical dopaminergic neurotransmission [113]. In this line, scientific evidence from brain imaging studies has demonstrated that carriers of the Val/Val genotype had lower “connection strength” between the hippocampus and the ventrolateral prefrontal cortex (VLPFC) while subjects performed a visual memory task [23]. On the other hand, primate studies have shown an increase in extracellular dopamine levels in the DLPFC during the performance of tasks that require the implementation of processes related to working memory, mainly in spatial tasks [114,115]. All this experimental evidence seems to confirm the importance of dopamine levels in the fronto-temporal network during cognitive processing [116], which could be proposed as a tentative explanation for the cognitive dysfunction shown by fibromyalgia patients in spatial, but not in verbal, working memory tests.

Finally, as the analysis carried out in the study showed, the influence of the COMT genotypes on the neuropsychological outcomes was independent of clinical symptoms (i.e., pain, depression and anxiety). Although some authors have found an influence of clinical outcomes on cognitive performance in fibromyalgia [84,86,117], others have failed to find this relationship [56,105,118,119,120]. Therefore, there is still some debate on this issue. Some authors have explained that clinical symptoms, especially pain, are not the origin of cognitive problems in fibromyalgia syndrome but are concomitant to these [119]. Although these symptoms (depression or anxiety) may contribute somewhat to cognitive dysfunction in fibromyalgia, they cannot explain all the deterioration that patients show in different cognitive domains [105]. This author has argued that even more research is needed to explore the role of clinical symptoms in cognition, as well as to provide more information about the combined effects of fibromyalgia with clinical symptoms on cognitive function.

Some limitations related to other potential influences should be taken into account for the interpretation of the current findings. Firstly, although the sample size was sufficient to obtain high statistical power, future studies should confirm these conclusions using larger samples. Secondly, the effects of dopamine on working memory are paradoxical [13,121], with opposite effects observed in similar studies. For this reason, the results derived from this study should be taken with caution since it has been observed that the relationship between the COMT genotypes and working memory not only depends on dopamine degradation but also on other genes related to dopamine transporters [29] or dopamine receptors [8,9,11,12,20]. These genes have been considered relevant for the modulation of dopamine transmission within deeper neural structures involved in working memory processing, such as the striatum or the PFC [1,122]. On the other hand, although the single SNP analysis has provided abundant evidence regarding the role of the COMT gene in working memory functioning, the use of a haplotype-based approach could complement this analysis due to its better sensitivity in representing the subjects’ availability of dopamine. Finally, although drug consumption affecting cognition was statistically controlled to neutralise its influence, future research only including fibromyalgia patients who are free of medication is recommended.

## 5. Conclusions

In summary, our data are the first to establish a role of COMT, a gene involved in the availability of dopamine, on working memory dysfunction in fibromyalgia patients. Specific genotypic variations of the COMT gene (i.e., Val/Val allele carriers) might contribute to diminished cognitive performance in fibromyalgia, mainly in visuo-spatial working memory tasks. These promising data could help to better characterize and detect working memory dysfunction in the more cognitively impaired patients and contribute to the design of more adjusted cognitive treatments for these chronic pain patients. Nevertheless, future studies should be done using a COMT haplotype-based approach to explore its influence on the working memory impairment in fibromyalgia in more depth, not only by means of neuropsychological assessments but also focusing on the neural indices underlying this cognitive dysfunction.

## Figures and Tables

**Figure 1 jcm-09-02479-f001:**
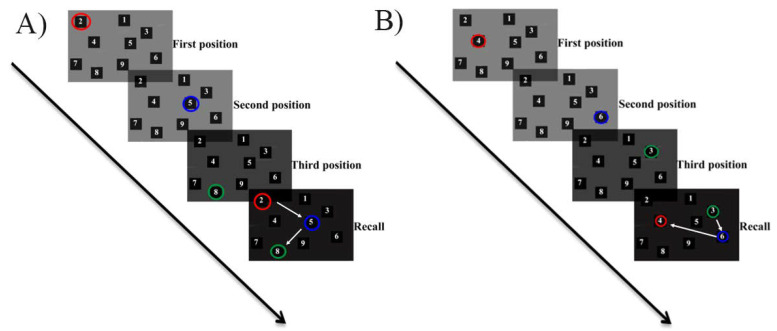
Spatial Span Test. Example of a trial representing a sequence length of three cubes. (**A**) Trial of the Spatial Span Test in the forward application; (**B**) Trial of the Spatial Span Test in the backward application. White arrows mark the correct order in which subjects should have remembered the tapping sequence.

**Figure 2 jcm-09-02479-f002:**
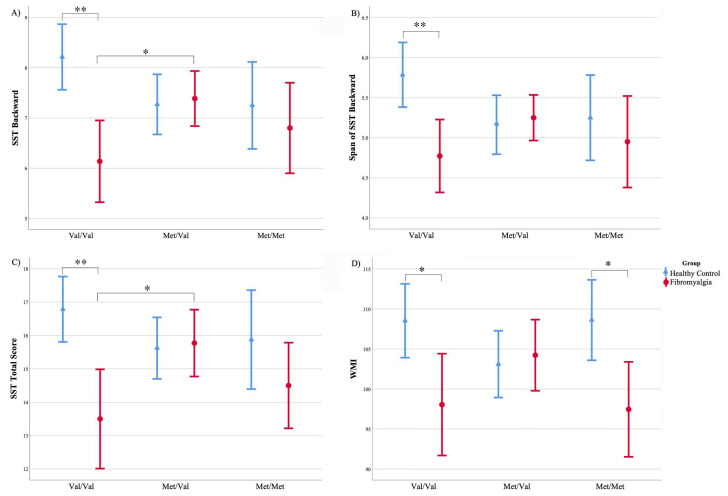
Means and standard errors related to the performance in the working memory tests. Data of fibromyalgia patients (red) and healthy controls (blue) are represented for each COMT genotype: Val/Val, Met/Val and Met/Met. (**A**) SST backward score, (**B**) span SST backward, (**C**) SST total score and (**D**) WMI. * *p* < 0.05; ** *p* < 0.001. SST, Spatial Span Test; WMI, Working Memory Index; Val, valine; Met, methionine.

**Table 1 jcm-09-02479-t001:** Means and standard deviations of age, levels of trait and state anxiety, depression, pain, drug consumption and educational levels for each group of participants. *p*-values for each statistical contrast are also included.

Clinical Variables	Fibromyalgia Patients	Healthy Control	*p*-Value
Age	51.38 (7.43)	51.42 (7.51)	0.975
STAI			
STAI-Trait	65.23 (28.13)	34.60 (26.67)	0.001
STAI-State	43.27 (26.70)	29.53 (25.49)	0.001
BDI	18.69 (9.28)	6.47 (4.86)	0.001
VAS Pain	6.26 (2.08)	0.977 (1.69)	0.001
FIQ-S	55.89 (18.09)	-	-
Drug consumption			
Antidepressants (%)	53.5	0.02	0.001
Analgesics (%)	59.30	0.02	0.001
Benzodiazepines (%)	33.7	0.02	0.001
Others (%)	50	25.93	0.001
Educational level			
Elementary studies (%)	16.28	16.05	0.347
Middle level (%)	60.47	50.62	
Superior university studies (%)	23.25	33.33	

STAI, State-Trait Anxiety Inventory; BDI, Beck Depression Inventory; VAS, visual analogue scale; FIQ-S, Spanish version of the Fibromyalgia Impact Questionnaire.

**Table 2 jcm-09-02479-t002:** Allele and genotype frequencies of COMT in fibromyalgia patients and healthy control group.

	Genotype Frequencies *n* (%)			Allele Frequencies	
Genotypes	HC (*n* = 81)	Fibromyalgia (*n* = 86)		HC (*n* = 81)	Fibromyalgia (*n* = 86)
Val/Val	28 (19.8)	22 (25.6)	Val	0.49	0.43
Met/Val	37 (45.7)	44 (51.2)	Met	0.51	0.57
Met/Met	16 (34.6)	20 (23.3)			

HC, healthy control participants; Val, valine; Met, methionine.

**Table 3 jcm-09-02479-t003:** Mean scores and standard deviations (in parenthesis) for working memory measures. Data are separated by group (fibromyalgia and HC) and genotype (Val/Val, Met/Val and Met/Met). *p*-values for the interaction between the COMT genotypes and group are also included.

	COMT						*p*-Value
	Healthy Control			Fibromyalgia			
Test	Val/Val	Met/Val	Met/Met	Val/Val	Met/Val	Met/Met	
Span of LNST	5.04 (0.88)	5.08 (0.89)	5.38 (0.62)	4.77 (1.11)	5.09 (1.15)	4.80 (0.91)	0.327
LNST score	10 (2.35)	10.05 (2.38)	10.75 (1.34)	9.05 (2.75)	9.86 (2.86)	8.75 (2.26)	0.191
Span of SST forward	5.79 (0.78)	5.89 (0.77)	5.69 (0.87)	5.14 (1.20)	5.70 (1.01)	5.30 (1.12)	0.412
SST forward score	8.57 (1.42)	8.43 (1.55)	8.63 (1.66)	7.36 (1.98))	8.39 (1.88)	7.70 (1.83)	0.147
Span of SST backward	5.79 (1.97)	5.16 (1.11)	5.35 (1.06)	4.77 (1.06)	5.25 (0.94)	4.95 (1.27)	0.020
SST backward score	8.21 (1.72)	7.27 (1.82)	7.25 (1.73)	6.14 (1.91)	7.39 (1.82)	6.80 (2.01)	0.005
SST total score	16.79 (2.58)	15.62 (3.32)	15.87 (2.96)	13.50 (3.48)	15.77 (3.32)	14.50 (2.87)	0.008
WMI	108.63 (10.06)	103.11 (12.69)	108.54 (12.21)	98.92 (14.95)	104.23 (14.75)	97.45 (13.29)	0.018

LNST, Letter and Number Sequence Test; SST, Spatial Span Test; WMI, Working Memory Index.

**Table 4 jcm-09-02479-t004:** Pearson correlations between the clinical questionnaire scores and neuropsychological measures.

	State Anxiety	Trait Anxiety	Depression	Pain
Span of LNST	−0.057	−0.051	−0.093	−0.068
LNST score	−0.087	−0.095	−0.157 *	−0.137
Span of SST forward	−0.083	−0.219 **	−0.129	−0.157 *
SST forward score	−0.116	−0.209 **	−0.134	−0.121
Span of SST backward	−0.143	−0.206 **	−0.150	−0.119
SST backward score	−0.085	−0.133	−0.130	−0.146
SST total score	−0.114	0.192*	−0.148	−0.149
WMI	−0.012	−0.131	−0.166 *	−0.117

* *p* < 0.05; ** *p* < 0.01.

**Table 5 jcm-09-02479-t005:** Means and standard deviations (in parenthesis) of working memory outcomes for fibromyalgia patients who were taking or not taking medications (benzodiazepines and antidepressants). *p*-values of the one way-ANOVAs are also included.

	Antidepressant ANOVAs			Benzodiazepine ANOVAs		
	Medication	No Medication	*p*-Value	Medication	No Medication	*p*-Value
Span of LNST	4.98 (0.97)	4.90 (1.236)	0.744	4.83 (0.92)	5.00 (1.18)	0.495
LNST score	9.43 (2.50)	9.35 (2.99)	0.887	8.97 (2.42)	9.61 (2.85)	0.299
Span of SST forward	5.61 (0.97)	5.30 (1.22)	0.197	5.59 (1.01)	5.40 (1.14)	0.471
SST forward score	8.24 (1.66)	7.65 (2.17)	0.159	8.14 (1.66)	7.88 (2.06)	0.557
Span of SST backward	5.15 (1.13)	4.95 (0.98)	0.384	5.14 (0.95)	5.02 (1.12)	0.624
SST backward score	7.17 (1.99)	6.65 (1.86)	0.213	7.10 (1.67)	6.84 (2.06)	0.558
SST total score	15.41 (2.98)	14.30 (3.71)	0.127	15.24 (2.98)	14.72 (3.56)	0.500
WMI	101.63 (13.31)	100.43 (16.25)	0.706	100.31 (13.29)	101.46 (15.43)	0.734

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
