# Peer review of "Effects of COMT Genotypes on Working Memory Performance in Fibromyalgia Patients"

_jcm, 2020, doi:10.3390/jcm9082479_

Round 1

Reviewer 1 Report

It is a good manuscript. To my mind, the main strength of the study is its originality, because it seems to be the first article which shows the relation between the COMT gene and working memory dysfunction in Fibromyalgia. Additionally, there aren´t objective errors in the methods or results sections. Moreover, the conclusions are supported by the presented data and previous findings. However, I recommend some aspects in order to ensure conceptual and methodological quality of the manuscript:

Introduction

  • Line 35: please add some examples of dopamine agonist drugs.
  • Lines 60-66: In the introductive section the authors described the main topic of the present study in a systematic and articulate way. However, when the authors describe cognitive deficits in Fibromyalgia a mention should be made to “fibro-fog”. Furthermore, it is now well recognized that among the higher cognitive domains impaired in Fibromyalgia there is “Theory of Mind”. A mention should be made (e.g., Di Tella, M., Castelli, L., Colonna, F., Fusaro, E., Torta, R., Ardito, R. B., et al. (2015) Theory of Mind and Emotional Functioning in Fibromyalgia Syndrome: An Investigation of the Relationship between Social Cognition and Executive Function. PLoS ONE, 10, http://dx.doi.org/10.1371/journal.pone.0116542)

Materials and Method

  • Line 99: Please, specify the meaning of the abbreviations AFYSYFACRO and AFAP.
  • Line 116: Please, add the code or reference of the Ethic Committee approval.
  • Line 300: It would be recommended to add after Table 4 a note or legend including the meaning of * and **
  • Authors should report the Cronbach’s alpha (internal consistency) for all the measures they used. Moreover, they have to explain if they calculated these values on the current data or took them from literature.
  • Authors should specify for each questionnaire the original version and the Spanish or validated version in the cases in which they used them.
  • It would be better if authors provided the score range for all questionnaires (minimum and maximum scores).
  • This section has a good structure and it has been well explained.
  • Were medications restricted or recorded prior to the experimental testing with saliva?
  • It would be advisable to include a section referred to the procedure.

Discussion

  • Have authors consider the possible bias related to self-questionnaires in Fibromyalgia? Please comment your ideas at this regard. For more information about this bias and the possibility to increase the intensity and frequency of self-reported symptoms due to negative affectivity in Fibromyalgia, see:
  1. Finan, P. H., Zautra, A. J., & Davis, M. C. (2009). Daily affect relations in fibromyalgia patients reveal positive affective disturbance. Psychosomatic Medicine, 71(4), 474-482. http://dx.doi.org/10.1097/PSY.0b013e31819e0a8b
  2. Kamping, S., Bomba, I. C, Kanske, P., Diesch, E., & Flor, H. Deficient modulation of pain by a positive emotional context in fibromyalgia patients. (2013) P 154(9), 1846-1855. http://dx.doi.org/10.1016/j.pain.2013.06.003
  3. Watson, D., & Pennebaker, J. W. (1989). Health complaints, stress and distress: Exploring the central role of negative affectivity. Psychological Review, 96, 234-254.
  • The discussion has been well-written and include the most important aspects.
  • Please, add the main limitations of the study.

Data availability statement

  • To enhance the reproducibility of your results, we recommend that if applicable your data were available in any public repository, such as OSFHOME or something similar, to allow the study replication. If it is not possible maybe authors might include a statement refer to the possibility to contact to them if any reader need the data.

Author Response

Reviewer 1

It is a good manuscript. To my mind, the main strength of the study is its originality, because it seems to be the first article which shows the relation between the COMT gene and working memory dysfunction in Fibromyalgia. Additionally, there aren´t objective errors in the methods or results sections. Moreover, the conclusions are supported by the presented data and previous findings. However, I recommend some aspects in order to ensure conceptual and methodological quality of the manuscript:

Introduction

  • Line 35: please add some examples of dopamine agonist drugs.

Response: Thank you for the suggestion. We have added some examples of dopamine agonist drugs (pergolide or bromocriptine) (see page 1, line 35).

  • Lines 60-66: In the introductive section the authors described the main topic of the present study in a systematic and articulate way. However, when the authors describe cognitive deficits in Fibromyalgia a mention should be made to “fibro-fog”. Furthermore, it is now well recognized that among the higher cognitive domains impaired in Fibromyalgia there is “Theory of Mind”. A mention should be made (e.g., Di Tella, M., Castelli, L., Colonna, F., Fusaro, E., Torta, R., Ardito, R. B., et al. (2015) Theory of Mind and Emotional Functioning in Fibromyalgia Syndrome: An Investigation of the Relationship between Social Cognition and Executive Function. PLoS ONE, 10,http://dx.doi.org/10.1371/journal.pone.0116542)

Response: Thank you your four helpful suggestion. We have introduced some information about the term fibrofog (see page 2, lines 61-62). Furthermore, following the reviewer recommendation, we have added a mention related to impairments in theory of mind, as a part of the most affected cognitive domains in fibromyalgia disease (line 66)

Materials and Method

  • Line 99: Please, specify the meaning of the abbreviations AFYSYFACRO and AFAP.

Response: Following the reviewer´s recommendation we have specified the meaning of the AFYSYFACRO and AFAP abbreviations (see page 3, line 101-102).

  • Line 116: Please, add the code or reference of the Ethic Committee approval.

Response: We upload the approval report of the ethics committee referring to this project and we added the reference code on the main text.

  • Line 300: It would be recommended to add after Table 4 a note or legend including the meaning of * and **

Response: Thank you for your comment. We have added a note explaining the de meaning of * and ** (see page 10, line 321)

  • Authors should report the Cronbach’s alpha (internal consistency) for all the measures they used. Moreover, they have to explain if they calculated these values on the current data or took them from literature.

Response: Thank you for your comment. We have reported the Cronbach´s alfa of all measures. We took these statistics from the literature, as explain in the manuscript.

  • Authors should specify for each questionnaire the original version and the Spanish or validated version in the cases in which they used them.

Response: Thank you for your suggestion. In all cases, the Spanish validated version of questionnaires was applied to the whole sample of participants. we have included this clarification in the description of the self-report questionnaires.

  • It would be better if authors provided the score range for all questionnaires (minimum and maximum scores).

Response: Thank you for your suggestion. We have added the score ranges of all the tests and questionnaires used, both clinical and neuropsychological (see Methods section lines 126-198)

  • This section has a good structure and it has been well explained.

  • Were medications restricted or recorded prior to the experimental testing with saliva?

Response: Indeed, during the study, the sociodemographic data were first collected, including the consumption of drugs (medication, dosage and the specialist who had prescribed it). Subsequently, the neuropsychological tests were carried out and finally, before completing the study, the saliva sample was collected. This information has been included in the procedure section (see pages 3, line 118-125).

  • It would be advisable to include a section referred to the procedure.

Response: Thank you for helpful suggestion. We have included a procedure section (see pages 3, line 118-125)

Discussion

  • Have authors consider the possible bias related to self-questionnaires in Fibromyalgia? Please comment your ideas at this regard. For more information about this bias and the possibility to increase the intensity and frequency of self-reported symptoms due to negative affectivity in Fibromyalgia, see:
  1. Finan, P. H., Zautra, A. J., & Davis, M. C. (2009). Daily affect relations in fibromyalgia patients reveal positive affective disturbance. Psychosomatic Medicine, 71(4), 474-482. http://dx.doi.org/10.1097/PSY.0b013e31819e0a8b
  2. Kamping, S., Bomba, I. C, Kanske, P., Diesch, E., & Flor, H. Deficient modulation of pain by a positive emotional context in fibromyalgia patients. (2013) P 154(9), 1846-1855. http://dx.doi.org/10.1016/j.pain.2013.06.003
  3. Watson, D., & Pennebaker, J. W. (1989). Health complaints, stress and distress: Exploring the central role of negative affectivity. Psychological Review, 96, 234-254.

Response: Thank you for your interesting comment. Self-reported questionnaires have been widely used in fibromyalgia to assess clinical symptoms of the disease. Although it is true, as the reviewer comments, the use of these instruments in fibromyalgia has some drawbacks. On the one hand, the presence of alexithymia (i.e., difficulty in identifying one's emotions) has been consistently reported in these patients (Sayar, Gulec, & Topbas, 2004; Tuzer et al., 2011). The inability to regulate emotions, especially the negative ones, increases the negative affect and has been associated with a worsening of symptoms such as pain (Di Tella et al., 2017). The presence of alexithymia in these patients is a relevant element in the use of self-report questionnaires since it can hinder the correct self-evaluation of the associated affective symptoms (Parling, Mortazavi, & Ghaderi, 2010).To solve this question, the used instruments based on structured interviews that has been postulated to complement the self-reported measures (Di Tella et al., 2017). Despite all this, the most relevant and novel results of the present study are those in which cognitive functioning has been measured. In this sense, the instruments used to measure working memory (i.e., WMS-III) are not self-reported, but have been administered by an expert psychologist (as it has been included in the procedure section) and the biases associated with the measurement of these variables have been reduced.

  • The discussion has been well-written and include the most important aspects.

  • Please, add the main limitations of the study.

Response: Thank you for your suggestion. A considerations or limitations section had been included in the manuscript. This section has been expanded following the comments of the reviewers (page 13)

Data availability statement

  • To enhance the reproducibility of your results, we recommend that if applicable your data were available in any public repository, such as OSFHOME or something similar, to allow the study replication. If it is not possible maybe authors might include a statement refer to the possibility to contact to them if any reader need the data.

Response: Thank you for your suggestion. The data of this study can be consulted in the following repository: 10.17605/OSF.IO/SHMB7. We have also included a section at the end of the manuscript including this information.

Reviewer 2 Report

The study explored the link between the COMT gene and working memory dysfunction in a sample of patients with fibromyalgia, supporting the idea that higher COMT enzyme activity could contribute to a more severe working memory impairment in this kind of pathology. In my opinion, the goal and implications of the study are of great relevance, especially in their effort to better understand the mechanisms underlying the cognitive impairment in fibromyalgia.

The manuscript is well written and contains all the required sections. The hypotheses are clearly stated and the study design is correct. The measures used in the study are adequate. The manuscript gives almost complete documentation that allows other researchers for a precise replication of the study. The choice of specific data analyses is adequate.

The presentation of arguments and results in the text is complete.

Despite that, I would like to suggest some minor revisions to improve the quality of the paper.

As the authors correctly explained, psychological variables could influence their statistical results, especially concerning the neuropsychological outcomes. Despite this, they not discussed their findings in the discussion section. Since regarding anxiety and depressive symptoms, studies in the literature have shown undefined results I would ask the authors to discuss these results with current literature data. Also, it should not be forgotten that a large percentage of patients with fibromyalgia reported comorbidities with personality disorders (Thieme et al., 2004; Rose et al., 2009; Uguz et al., 2010; Gumà-Uriel et al. 2016; Kayhan et al., 2016). In the participants' section, the authors stated that the diagnosis of fibromyalgia was carried out by different rheumatologists, but they did not mention the exclusion or inclusion of psychiatric disorders among patients (and/or among healthy subjects). Has this factor been controlled?

Finally, I would ask the authors to include the relatively small sample size as a limitation.

The discussion is poorly structured in terms of clinical comment on the results obtained and it is unclear what this study adds to the literature: results need to be better discussed in accordance to the relevant literature.

Author Response

Reviewer 2

The study explored the link between the COMT gene and working memory dysfunction in a sample of patients with fibromyalgia, supporting the idea that higher COMT enzyme activity could contribute to a more severe working memory impairment in this kind of pathology. In my opinion, the goal and implications of the study are of great relevance, especially in their effort to better understand the mechanisms underlying the cognitive impairment in fibromyalgia.

The manuscript is well written and contains all the required sections. The hypotheses are clearly stated and the study design is correct. The measures used in the study are adequate. The manuscript gives almost complete documentation that allows other researchers for a precise replication of the study. The choice of specific data analyses is adequate.

The presentation of arguments and results in the text is complete.

Despite that, I would like to suggest some minor revisions to improve the quality of the paper.

  • As the authors correctly explained, psychological variables could influence their statistical results, especially concerning the neuropsychological outcomes. Despite this, they not discussed their findings in the discussion section. Since regarding anxiety and depressive symptoms, studies in the literature have shown undefined results I would ask the authors to discuss these results with current literature data.
  • The discussion is poorly structured in terms of clinical comment on the results obtained and it is unclear what this study adds to the literature: results need to be better discussed in accordance to the relevant literature

Response: Thank you for the reviewer’s suggestions. We have included the discussion about the relationship between clinical symptoms and neuropsychological outcomes (see pages 12-13, lines 449-459). In this part we highlight the presence of disparate literature on the relationship between these variables in fibromyalgia and the need to continue researching in order to better understand this topic.

  • Also, it should not be forgotten that a large percentage of patients with fibromyalgia reported comorbidities with personality disorders (Thieme et al., 2004; Rose et al., 2009; Uguz et al., 2010; Gumà-Uriel et al. 2016; Kayhan et al., 2016). In the participants' section, the authors stated that the diagnosis of fibromyalgia was carried out by different rheumatologists, but they did not mention the exclusion or inclusion of psychiatric disorders among patients (and/or among healthy subjects). Has this factor been controlled?

Response: Thank you for your comment. The patients had a main diagnosis of fibromyalgia evaluated by a rheumatologist, in addition, both healthy participants and patients, should not have a diagnosis of psychiatric (including personality disorder) or neurological disorder. On page 3, lines 113-114, you can read these criteria used.

  • Finally, I would ask the authors to include the relatively small sample size as a limitation.

Response: Thank you for your suggestion. We have included this limitation in the limitation section of the manuscript. We have specifically included in the manuscript the exclusion of subjects diagnosed with personality disorder (page 13, lines 461-462)